# Treatment of Marmoset Intracerebral Hemorrhage with Humanized Anti-HMGB1 mAb

**DOI:** 10.3390/cells11192970

**Published:** 2022-09-23

**Authors:** Dengli Wang, Daiki Ousaka, Handong Qiao, Ziyi Wang, Kun Zhao, Shangze Gao, Keyue Liu, Kiyoshi Teshigawara, Kenzo Takada, Masahiro Nishibori

**Affiliations:** 1Department of Pharmacology, Faculty of Medicine, Dentistry, and Pharmaceutical Sciences, Okayama University, Okayama 7008558, Japan; 2Research Fellow of Japan Society for the Promotion of Science, Tokyo 1020083, Japan; 3Department of Molecular Biology and Biochemistry, Faculty of Medicine, Dentistry and Pharmaceutical Sciences, Okayama University, Okayama 7008558, Japan; 4School of Pharmaceutical Sciences, Tsinghua University, Beijing 100084, China; 5Sapporo Laboratory, EVEC, Inc., Sapporo 0606642, Japan; 6Department of Translational Research and Drug Development, Faculty of Medicine, Dentistry and Pharmaceutical Sciences, Okayama University, Okayama 7008558, Japan

**Keywords:** intracerebral hemorrhage, HMGB1, antibody therapy, non-human primate

## Abstract

Intracerebral hemorrhage (ICH) is recognized as a severe clinical problem lacking effective treatment. High mobility group box-1 (HMGB1) exhibits inflammatory cytokine-like activity once released into the extracellular space from the nuclei. We previously demonstrated that intravenous injection of rat anti-HMGB1 monoclonal antibody (mAb) remarkably ameliorated brain injury in a rat ICH model. Therefore, we developed a humanized anti-HMGB1 mAb (OKY001) for clinical use. The present study examined whether and how the humanized anti-HMGB1 mAb ameliorates ICH injury in common marmosets. The results show that administration of humanized anti-HMGB1 mAb inhibited HMGB1 release from the brain into plasma, in association with a decrease of 4-hydroxynonenal (4-HNE) accumulation and a decrease in cerebral iron deposition. In addition, humanized anti-HMGB1 mAb treatment resulted in a reduction in brain injury volume at 12 d after ICH induction. Our in vitro experiment showed that recombinant HMGB1 inhibited hemoglobin uptake by macrophages through CD163 in the presence of haptoglobin, suggesting that the release of excess HMGB1 from the brain may induce a delay in hemoglobin scavenging, thereby allowing the toxic effects of hemoglobin, heme, and Fe^2+^ to persist. Finally, humanized anti-HMGB1 mAb reduced body weight loss and improved behavioral performance after ICH. Taken together, these results suggest that intravenous injection of humanized anti-HMGB1 mAb has potential as a novel therapeutic strategy for ICH.

## 1. Introduction

Spontaneous intracerebral hemorrhage (ICH) remains a significant cause of mortality and morbidity [1]. In regard to treatment for ICH, most class I evidence is related to supportive care measures, such as blood pressure control, coagulopathy management, and glucose management [2]. No FDA-approved treatments or therapies have been established for enhancing post-ICH recovery [2]. After ICH, blood extravasation may release thrombin, fibrinogen, hemoglobin (Hb), and Hb degradation products, and the toxic effects of these proteins on brain cells can trigger secondary brain injury. For decades, the clinical trials of ICH have mainly focused on such secondary injury, addressing such topics as minimally invasive surgical treatment, hematoma clearance, iron chelation, platelet dysfunction, and epilepsy [3]. However, no specific treatment showed effectiveness on clinical outcomes in these clinical trials [2]. For example, minimally invasive surgery plus recombinant tissue plasminogen activator (rt-PA) for ICH evacuation (MISTIE) did not improve the functional outcome for patients at 365 days after moderate to large ICH in a phase III clinical trial [4]. Deferoxamine, an iron chelator, did not significantly improve clinical outcomes after ICH in a phase II clinical trial [5]. Although desmopressin improved platelet activity, there are no data on its potential effects on reducing hematoma growth [6]. None of these measures have been translated to clinical use, even though they have good effects in rodent models [1]. For this reason, we conducted ICH experiments in a non-human primate that possesses a higher-order brain similar to humans. 

High mobility group box1 (HMGB1), one of the damage-associated molecular patterns (DAMPs), is known to initiate inflammation once released into the extracellular space from cellular nuclei [7]. Released HMGB1 stimulates toll-like receptor 4/2 (TLR4/2) or the receptor of AGE (RAGE) to trigger the expression of inflammatory cytokines by the NF-κB and MAP kinase signaling pathway [8]. In addition, HMGB1 forms complexes with IL-1β and CXCL12, leading to enhancement of the affinity for the respective cognate receptors [9]. It has also been well established that HMGB1 plays a critical role in the pathogenesis of central nervous system (CNS) diseases, including ischemic brain infarction, ICH, traumatic brain injury (TBI), neuropathic pain, epilepsy, and Parkinson’s disease [10,11,12,13,14,15,16]. In these CNS diseases, disruption of the blood-brain barrier (BBB) facilitated brain inflammation by the leakage of plasma factors and induction of immune cell migration, both of which were mediated by HMGB1 [9]. 

Following ICH, the toxic effects of Hb, heme, iron, and other blood components on the surrounding brain contribute to ICH-induced secondary injury [17]. These blood components all play an essential role in hemorrhage-induced BBB dysfunction, resulting in brain edema formation and neuronal damage [18]. Heme or iron-mediated free radical production is one of the causes of neurotoxicity [19]. Hb functions synergistically with HMGB1 to promote the release of proinflammatory cytokines from macrophages through Myd88- and TRIF-independent pathways [20]. HMGB1 was also shown to cooperatively enhance thrombin-induced coagulation and inflammation, leading to multiple organ failures in a rat model of disseminated intravascular coagulation [21]. Our previous research indicated that HMGB1 plays a critical role in the development of ICH-induced secondary injury through the amplification of plural inflammatory responses in rats [11], whereas a rat anti-HMGB1 mAb has been shown to remarkably ameliorate ICH injury in rats, and this amelioration was associated with a decrease in activated microglia and astrocytes and a suppression of the expression of inflammation-related factors [11]. Other researchers reported that the HMGB1 inhibitors glycyrrhizin and ethyl pyruvate (EP) reduced brain edema and improved the functional outcome in a rat ICH model by interfering with the binding of HMGB1 with TLR4 or RAGE [22,23,24]. Our previous study showed that a rat anti-HMGB1 mAb had neuroprotective effects during the acute phase of ICH [11]. However, it is also well known that a promising candidate drug screened by a mouse or rat model was not beneficial in humans [1,25]. Therefore, in the present study, we explored the prolonged effects of a humanized anti-HMGB1 mAb in a marmoset model of ICH since primates and humans are genetically closer relatives than rodents and humans. Neurobehavioral outcomes and body weight were measured throughout a 12-day observation period to evaluate the general condition of both animals treated with normal human immunoglobulin G (IgG) as a control (IgG control group) and those treated with anti-HMGB1 mAb (anti-HMGB1 mAb group). We observed that the humanized anti-HMGB1 mAb exerted both acute and long-lasting beneficial effects on marmoset ICH by reducing deleterious proinflammatory responses and promoting hematoma absorption. As a result, the humanized anti-HMGB1 mAb ameliorated secondary brain damage. The humanized anti-HMGB1 mAb treatment may provide a novel therapy for human ICH.

## 2. Materials and Methods

### 2.1. Animals

Common marmosets (C. jacchus) were bred for this study at the Department of Animal Resources Advanced Science Research Center, Okayama University. The animals were pair-housed, and husbandry and veterinary services were provided. Both male and female marmosets of 2–4 years of age (body weight, 292–356 g) were used in this study. The total number of marmosets was 19. Control IgG group included 5 males and 4 females and α-HMGB1-treated group included 3 males and 3 females. One female was used for a sham control without ICH. Three males were used for the observations after ICH without any treatment. All procedures for these experiments were approved by the Committee on Animal Studies of Okayama University and carried out following the guidelines of Okayama University for animal studies. The room was maintained at a constant temperature (27 °C) and relative humidity (50%) on a 12:12-h light/dark cycle. 

### 2.2. Production of Humanized Anti-HMGB1 Monoclonal Antibody

The expression plasmid vector of recombinant anti-HMGB1 mAb (OKY001) was produced by insertion of the CDR sequence of the original rat anti-HMGB1 mAb (#10-22) into the human IgG frame. The construct was transfected into CHO cells, and the culture was continued for 7 d at 37 °C. The cell suspensions were centrifuged, and the resultant supernatant was used as the anti-HMGB1 mAb in this study. The purification steps consisted of affinity purification by protein G and anion exchange chromatographies. After dialysis against PBS, the purified samples were used for analysis and animal experiments. 

### 2.3. Western Blot Analysis

Different amounts of recombinant human HMGB1 (Abnova, Taipei, Taiwan) or native calf HMGB1/2 (Wako Pure Chemical, Osaka, Japan) were electrophoresed on SDS-PAGE. After transfer onto the PVDF membrane, HMGB1 was detected by humanized anti-HMGB1 mAb (OKY-001) or rat anti-HMGB1 mAb (#10-22) at different concentrations and followed with HRP-conjugated secondary antibodies.

### 2.4. ICH Induction and the Treatment of Marmosets

ICH was performed as described previously [11] with some modifications. Marmosets were placed on a heating pad and secured into a stereotactic frame after anesthesia with 1–3% isoflurane (Pfizer, New York, NY, USA). Under sterile conditions, a 30-gauge needle was inserted into the right striatum at a position 5.4 mm anterior, 3.5 mm lateral, and 8.3 ventral from the bregma. Collagenase IV (0.18 U) in 12 μL saline was delivered at a constant rate of 2.5 µL/min to establish the ICH model. Humanized anti-HMGB1 mAb (OKY001) or normal human IgG (IgG from human serum, Sigma-Aldrich, Tokyo, Japan) as a control was administered through the tail vein 10 min after ICH induction. The microinjection needle was left in place for an additional 5 min to prevent backflow. After microinjection, we removed the needle, filled the burr hole with bone wax, and sutured the wound. After recovery from anesthesia, marmosets were returned to their cages and provided free access to water and food. 

### 2.5. Immunohistochemistry Staining

For immunohistochemical staining, the paraffin-embedded brain sections were prepared at the indicated time points. The primary antibodies used in the experiments were anti-HMGB1 Ab (R&D Systems Inc., Minneapolis, MN, USA), anti-4-HNE Ab (Abcam Plc, Cambridge, UK), anti-neuronal nuclear protein (NeuN) Ab (Abcam Plc, Cambridge, UK), anti-microtubule associated protein 2 (MAP2) Ab (Abcam Plc, Cambridge, UK), anti-ionized calcium-binding adaptor molecule 1 (Iba1) Ab (Wako, Osaka, Japan), anti-glial fibrillary acid protein (GFAP) Ab (Abcam Plc, Cambridge, UK), anti-cluster of differentiation 31 (CD31) Ab (Abcam Plc, Cambridge, UK) and anti-PDGFRβ (Abcam Plc, Cambridge, UK). To examine the cellular source and the translocation of HMGB1, double immunohistochemical staining was performed using the cell marker antibodies MAP2, GFAP, and iba1 and an antibody against HMGB1. To investigate the cellular source and the localization of 4-HNE, double or triple immunohistochemical staining was performed with the cell marker antibodies NeuN, MAP2, GFAP, iba1, PDGFRβ, and CD31 and an antibody against 4-HNE. The sections were then incubated with secondary Abs labeled with Alexa-488, Alex647, Alex594, or Alexa-555. The secondary Abs were purchased from Invitrogen (Tokyo, Japan). After that, the sections were mounted using VECTASHIELD Hard Set Mounting Medium with DAPI (Vector Laboratories Inc., Burlingame, CA, USA) and observed under an LSM 780 confocal microscopy system (Carl Zeiss Inc., Jena, Germany). The counting of immune-positive cells was carried out in a blinded manner.

### 2.6. Enzyme-Linked Immunosorbent Assay of HMGB1 and 4-HNE Adduct

For the measurement of HMGB1 and 4-HNE adduct levels in plasma samples, blood samples were collected from the tail vein under anesthesia. The HMGB1 concentration was determined by using an HMGB1 ELISA kit (Sino-test Co., Sagamihara, Japan) in accordance with the manufacturer’s protocol. The plasma 4-HNE adduct levels were measured using an OxiSelect™ HNE adduct competitive ELISA kit (Cell Biolabs, San Diego, CA, USA). Fifty microliter plasma samples were diluted to 100 μL with PBS. The samples were then added to a 96-well protein binding plate and left at 4 °C for overnight. The absorbance was detected at 450 nm after incubation with secondary antibody-HRP-conjugate. The plasma 4-HNE adduct concentrations were calculated according to the standard curve [26].

### 2.7. TUNEL Staining

The marmoset brains were fixed with 4% PFA at 3, 7, and 12 d after ICH. The paraffin-embedded sections were processed for a terminal deoxynucleotidyl transferase-mediated dUTP-biotin nick end labeling assay (Takara, Tokyo, Japan) according to the manufacturer’s protocol. The brain sections were treated with proteinase K (20 μg/mL) at room temperature for 20 min, and then 50 μL of an FITC-labeling reaction mixture was added onto the slides and allowed to react for 90 min in a 37 °C humidified chamber. Finally, the sections were counterstained with DAPI. 

### 2.8. Prussian Blue Staining

Prussian blue staining was used to detect the presence of iron in the ferric state, which includes ferritin and hemosiderin, inside the cells [27] according to the manufacturer’s protocol. Briefly, deparaffinated sections were washed with distilled water and then incubated with Prussian blue stain (a mixture of equal volumes of potassium ferrocyanide solution and hydrochloric acid solution) for 3 min, then rewashed with distilled water and counterstained with hematoxylin for 5 min. The slides were observed under a bright-field microscope to determine the distribution of iron oxide inside the cell.

### 2.9. Real-Time PCR

Real-time PCR. In this study, we established an ICH model in rats and then treated the ICH rats with humanized anti-HMGB1 10 min after ICH induction. A quantitative real-time polymerase chain reaction (qRT-PCR) was performed as described previously [11]. The brain samples for qRT-PCR were obtained from peri-hematomal regions (each sample weighed about 25 mg). The primers used for the analysis of mRNA expression were from a previous study [11]. The expressions of the following molecules were determined: interleukin-1β (IL-1β), IL-6, inducible nitric oxide synthase (iNOS), IL-8R and glyceraldehyde-3-phosphate dehydrogenase (GAPDH). GAPDH expression was used as an internal control to normalize cDNA levels. The relative fold gene expression of samples was calculated with the 2^−ΔΔCT^ method.

### 2.10. Computed Tomographic (CT) Examination and CT Sata Analysis

Brain CT scans were performed using an X-ray CT system (Latheta LCT-200; Hitachi Aloka Medical, Tokyo, Japan). During the anesthesia, we injected 2.5 mL omnipaque (i.v.) in a single dose, followed by a continuous infusion rate of 10 mL/h. When the injection volume reached 6 mL, the marmoset was subjected to a 15 min CT scan.

The open-source software ITK-SNAP (Penn Image Computing and Science Laboratory, Pennsylvania, PA, USA) was used to measure the volumes of intracerebral hemorrhage-induced lesions with a semi-auto three-dimensional segmentation algorithm. Briefly, all CT images were imported as DICOM datasets and showed slices of three dimensions. After Hounsfield unit (HU) thresholding (upper, 500; lower, 120), 2–3 seed points were placed within the hemorrhagic lesions. A threshold-based region growing algorithm (1) was then used to segment the hemorrhagic lesions from normal cerebral tissue after 100–150 iterations (α = 1; β = 0.05; γ = 0). The number of iterations was determined according to the conditions of each sample. Finally, the intracerebral hemorrhage-induced lesions were further confirmed and modified using the “adaptive brush” tool in the ITK-SNAP software.

### 2.11. Evaluation of Neurological Function and Body Weight

To assess motor function (i.e., for behavioral testing), a grip strength test (GPM-100B; MELQUEST, Toyama, Japan) was performed before the injury and at 1, 2, 3, 6 and 12 d after brain injury. Briefly, the marmosets were held by the torso, and allowed to grip the grid of the device with the contralateral forelimb and hindlimb of the hemorrhage side. Then, the marmosets were gently pulled downwards, and the peak of the grip strength was recorded. Five trials were performed for each marmoset and the average value was used as the animal’s contralateral grip force at the indicated time point. We measured the body weight before ICH and 1, 2, 3, 6, and 12 d after brain injury using a balance scale (CS series; OHAUS) under inhalation anesthesia.

### 2.12. Cell Preparation and Flow Cytometry

Human monocytic THP-1 cells were purchased from RIKEN BioResource Center (BRC, Kyoto, Japan). The cells were maintained in RPM1-1640 supplemented with 10% FBS, 2 mM glutamine, and 1% penicillin- streptomycin solution in a 5% CO2 humidified atmosphere and passaged every 3 d. In all experiments, THP-1 cells were pre-stimulated by dexamethasone (250 nM) for 2 d to induce CD163 expression. Then, 5 × 10^5^ cells were added to each microtube, and the tubes were centrifuged at 3000 rpm for 10 min. After washing twice with PBS, the cells were divided into 2 groups as follows. (1) For the endocytosis experiment of Hb, cells were stimulated with Alexa-647-labeled human Hb (1 µM) and human haptoglobin (0.0001 µM, 0.001 µM, 0.01 µM, or 1 µM) for 15 min in an FBS-free medium. (2) For the endocytosis experiment of Hb in the presence of HMGB1, cells were stimulated with Alexa-647-labeled human Hb (1 µM) and human haptoglobin (1 µM) with recombinant human HMGB1 (0.5, 1, 2, 5, or 10 µg/mL). Then, the cells were washed twice with PBS and incubated with CD163 primary antibody (333605; Biolegend, San Diego, CA, USA) for 1 h, at 4 ℃. Finally, the wells were fixed with 0.5% PFA for MACSQuant analysis. Pearson’s Correlation Coefficient was calculated using the median value of the intensity of Hb in Hb^+^/CD163^+^ cells. 

### 2.13. Statistical Analysis

Data were expressed as means ± standard error of the mean (SEM), and statistical analysis was performed using Prism software. Statistical analysis was carried out by one-way ANOVA for multiple comparisons and 2-way ANOVA for the grip strength test analysis and body weight change analysis. The significance of differences in the brain injury volume between the IgG control and humanized anti-HMGB1-treated groups was determined using Student’s *t*-test. The levels of significance were set at *p* ≤ 0.05, *p* ≤ 0.001, *p* ≤ 0.05 and *p* ≤ 0.001. For flow cytometry analysis, the non-debris cell population was identified using a flexible statistical model-based clustering approach by flowClust (https://doi.org/10.1002/cyto.a.20531, accessed on 27 May 2022) R software. The gating threshold for Hb or Alexa-647 positive cells was defined based on the negative control group. The R script of flow cytometry analysis is available from the author’s GitHub (https://github.com/wong-ziyi/FlowCytometricAnalysisFrame, accessed on 27 May 2022).

Data availability. The data that support the findings of this study are available from the corresponding author upon reasonable request.

## 3. Results

### 3.1. Characterization of Humanized Anti-HMGB1 mAb (OKY-001)

The humanized anti-HMGB1 mAb (OKY-001) was purified from the supernatant of CHO cells using affinity and anion exchange chromatography. The mAb (OKY-001) recognized recombinant human HMGB1 produced in sf9 cells as well as bovine thymus-derived HMGB1 very efficiently with higher affinity than that of the original rat mAb (#10-22) in western blotting (Figure 1A,B). The detection efficiency of humanized anti-HMGB1 mAb (OKY-001) compared with the original rat mAb (#10-22) was also demonstrated by an Elisa assay (Appendix A)

### 3.2. Effect of Humanized Anti-HMGB1 mAb on HMGB1 Levels in the Plasma after ICH

The protocol for establishing collagenase-induced hemorrhage in marmosets is shown in Figure 1C. The measurement of plasma HMGB1 in marmosets of the IgG control group revealed a clear increase in HMGB1 levels at 24 h and 48 h after ICH compared with the sham controls. This increase was significantly inhibited by intravenous administration of humanized anti-HMGB1 mAb (2 mg/kg; Figure 1D). These results suggested that plasma HMGB1 levels were increased by ICH induction in marmosets, and humanized anti-HMGB1 neutralized almost all the upregulated HMGB1 as in the case of the rat hemorrhage model in our previous research [11].

### 3.3. Effect of Humanized Anti-HMGB1 mAb on 4-hydroxynonenal (4-HNE) Adduct Levels in the Plasma after ICH

As a representative oxidative stress biomarker, the levels of 4-HNE adduct were determined at 2, 7, and 12 d after ICH. 4-HNE adduct levels were increased significantly (by almost 10-fold) in the IgG control group at 2 d after ICH induction compared to the sham group. The levels of 4-HNE adduct gradually decreased up to 12 d after ICH. The treatment with humanized anti-HMGB1 significantly inhibited the increase in plasma 4-HNE adduct levels at 2 and 7 d after ICH (Figure 1E). The results indicated that anti-HMGB1 mAb inhibited oxidative stress, as represented by plasma 4-HNE adduct levels, during the subacute phase of ICH. Concerning the systemic response of WBC count, the acute increase in the Con-IgG group one day after ICH was significantly inhibited by anti-HMGB1 mAb treatment (Figure 1F).

### 3.4. Release of HMGB1 from Neurons, Astrocytes, and Microglia

The release of HMGB1 in the brain was observed by immunofluorescence. As shown in previous studies [11], in the sham group, HMGB1 was localized in the nuclear compartment of cells, including neurons, astrocytes and microglia (Figure 2A,C,E). In the peri-hematomal regions, the nuclear immunoreactivities of HMGB1 were decreased significantly or even disappeared in most of the MAP-2-positive neurons 3 days after ICH induction (Figure 2B). The disappearance of HMGB1 immunoreactivities was also observed in a portion of the GFAP-positive astrocytes (Figure 2D) and a few iba1-positive microglia (Figure 2F). The similar phenomenon was also observed even at 7 days after ICH (Appendix A).

### 3.5. Effects of Humanized Anti-HMGB1 mAb on Brain Injury

To evaluate brain injury, we performed brain CT examinations in the control and humanized anti-HMGB1 mAb-treated groups at 12 d after ICH induction. The whole volume of the injured area detected of contrast material by the infiltration with a clear boundary was measured by ITK-SNAP software. The typical CT images from two groups (the IgG control-treated and humanized anti-HMGB1 mAb-treated groups) are shown in Figure 3C. The quantitative measurement results are shown in Figure 3D. There was no leakage of contrast material except at the injured site. It can be clearly seen that anti-HMGB1 mAb treatment reduced the region of brain injury at 12 d after ICH.

### 3.6. Anti-HMGB1 mAb Improved Neurological Deficits and Ameliorated Body Weight Loss after ICH

Neurological function was evaluated using the grip strength test. On the first day after ICH induction, neurologic deficits were found in all marmosets. The decrease in grip strength in control marmosets was exacerbated up to 3 d after ICH. However, the marmosets treated with anti-HMGB1 mAb showed significant recovery from their initial deficits in the grip strength test early at 3 d after ICH induction (Figure 3A). We also measured the body weight to estimate the whole-body condition at each time point. The body weight of the IgG control-treated marmosets gradually decreased with the peak loss at 3 d after ICH. Even after 12 d, the body weight did not recover to the initial levels (Figure 3B). The body weights of the marmosets in the anti-HMGB1-treated group recovered to the initial levels at 6 d and thereafter (Figure 3B).

### 3.7. Anti-HMGB1 mAb Treatment Induced a Decrease in TUNEL-Positive Cells

We evaluated ICH-induced cell death at the border of the hematoma area using TUNEL staining. The number of TUNEL-positive cells increased and reached a peak at the hematoma border at 7 d after ICH (Figure 4A). There were still a large number of TUNEL-positive cells observed at 12 d after ICH. The number of TUNEL-positive cells in the brains of the anti-HMGB1 mAb-treated animals at 12 d was significantly lower than that in the IgG control-treatment group (Figure 4B,C).

### 3.8. Histological Studies on the Effects of Humanized Anti-HMGB1 mAb

Hematoxylin-eosin staining of brain sections from the IgG control-treatment marmosets revealed that there were diffuse hemosiderin depositions (brown depositions in Figure 4E) in the hematoma area at 12 d after ICH induction. In contrast, the amount of hemosiderin deposition was decreased in the same areas in the marmosets treated with humanized anti-HMGB1 mAb. Prussian blue staining was used to detect iron depositions in cells. In accordance with the results of HE staining, Prussian blue staining revealed that a large cluster of iron-positive cells was found at the border of the hematoma at 12 d after ICH induction. However, only a few of the blue-stained cells were observed in the same area in the anti-HMGB1 mAb-treated group (Figure 4F,G).

### 3.9. 4-HNE Induction and Localization in Hematoma and Peri-Hematoma Regions after ICH

Lipid peroxidation is one of the major manifestations of oxidative stress in the brain. 4-HNE, which is produced by peroxidation of unsaturated free fatty acids in the membrane, is considered to play a crucial role in the oxidative injury of biomolecules [28,29], including proteins, lipids, and nucleic acids. In the present study, 4-HNE formation was detected by immunohistochemical staining. The accumulation of 4-HNE was gradually increased in a time-dependent manner from 3–18 d after ICH induction (Figure 5A). When we focused on the 4-HNE immunoreactivities at 12 d after ICH, we found that the administration of anti-HMGB1 mAb resulted in a significantly lower level of 4-HNE accumulation in the brain compared with the brains of the IgG control-treated animals (Figure 5B). When we counted the numbers of positive cells in the indicated fields in both groups, 55% suppression was observed by the treatment with anti-HMGB1 mAb (Figure 5C). In addition, we observed that the 4-HNE immunoreactivities were restricted to activated microglia cells and neurons, and were not observed in astrocytes or pericytes within the hematoma or at the border of the hematoma 7 d after ICH (Figure 5D). In the peri-hematoma area, we frequently observed that 4-HNE immunoreactivities were restricted to Iba1^+^ cells, which might migrate and be present between astrocyte end feet and pericytes in the blood vessels (Figure 5E,F). However, there is no 4-HNE observed in contralateral side (Figure 5G). The characteristic distribution of 4-HNE-positive microglia suggested that 4-HNE, which accumulated in migrated microglia cells or macrophages, may be related to vascular events that lead to further inflammation in the brain (Figure 5E,F). 

### 3.10. Co-Localization of HMGB1 and 4-HNE

At 7 d after ICH, we rarely observed the co-localization of HMGB1 and 4-HNE (Figure 6A). The co-localization of HMGB1 and 4-HNE appeared clearly in DAPI-positive nuclei 12 d after ICH in the IgG control-treatment group (Figure 6B). However, such co-localization was less clearly observed in the anti-HMGB1 group (Figure 6B), which may indicate that, after ICH, 4-HNE or 4-HNE adduct diffuse within the cells have the possibility to form a complex with HMGB1 in the nuclei which damages the physiological function of HMGB1.

### 3.11. Effects of Humanized Anti-HMGB1 on the Expression of Inflammation-Related Molecules

To analyze the anti-inflammatory mechanism underlying the effects of humanized anti-HMGB1 mAb, we measured the expression of inflammation-related molecules in the peri-hematomal region of the striatum by quantitative real-time PCR in a rat ICH model. The expressions of IL-1β, IL-6, iNOS, and IL-8R were all upregulated on the ipsilateral side of the IgG control-treated rats compared with the sham group at 24 h after ICH (Appendix A). The upregulations of IL-1β, IL-6, and iNOS were significantly suppressed by around 50% by the treatment with humanized anti-HMGB1 mAb.

### 3.12. Inhibition of Haptoglobin-dependent Uptake of Hb by Recombinant HMGB1

Next, we established an in vitro cell assay system to analyze the effects of HMGB1 on haptoglobin-dependent uptake of Hb into macrophages. We assessed the uptake of fluorescence-labeled Hb (alex647-Hb) into THP-1 cells, which were pretreated with dexamethasone (250 nM) for 2 d to induce CD163 expression. Flow cytometry studies demonstrated that increasing doses of haptoglobin increased the uptake of Hb in THP-1 cells in a CD163-dependent manner (Figure 7A,C), which was consistent with a previous report [30]. To explore whether HMGB1 affects the CD163/haptoglobin-dependent Hb uptake in THP1 cells, we incubated dexamethasone-pretreated THP1 cells with alex647-Hb (1 µM), haptoglobin (1 µM) and increasing doses of HMGB1 (0.5, 1, 2, 5, and 10 µg/mL). About 21.39% of the THP1 cells stained positive after exposure to 1 µM Alex647-Hb and haptoglobin in the absence of HMGB1, while 5.75, 5.03, 3.25, 3.05, and 2.38% of cells stained positive after exposure to 1 µM Alex647-Hb and haptoglobin in the presence of 0.5 µg/mL, 1 µg/mL, 2 µg/mL, 5 µg/mL, and 10 µg/mL HMGB1, respectively (Figure 7B,D). The median intensity of Hb in Hb^+^/CD163^+^ cells was significantly correlated with the concentration of HMGB1 (Figure 7D). This suggested that the Hb uptake into macrophages mediated by CD163/haptoglobin was reduced in the presence of HMGB1 in a dose-dependent manner with a negative linear correlation.

## 4. Discussion

We previously demonstrated that intravenous injection of rat anti-HMGB1 mAb remarkably ameliorated brain injury in a rat ICH model induced by local injection of collagenase IV into the striatum [11]. Several pharmacological agents have been shown to have neuroprotective efficacy in rodent models but have exhibited little or no therapeutic effects in clinical trials [31]. To bridge this biological gap and provide a strategy for advancing ICH therapy, in this study, we modeled ICH using the common marmoset, a non-human primate that is genetically much closer to humans than rodents. We successfully developed a marmoset ICH model induced by local injection of a minute amount of collagenase IV into the striatum and tested the neuroprotection conferred by intravenous infusion of humanized anti-HMGB1 mAb. The results demonstrated that humanized anti-HMGB1 mAb effectively reduced the release of HMGB1 from the brain into plasma, and this reduction was associated with a decrease in 4-HNE adduct accumulation in the brain and plasma. In addition, treatment with humanized anti-HMGB1 mAb ameliorated the brain injury at 12 d after ICH induction. In vitro experiments showed that haptoglobin increased the CD163-dependent uptake of Hb by THP-1 cells pre-stimulated with dexamethasone. However, recombinant HMGB1 inhibited the uptake of Hb by THP-1 cells in the presence of haptoglobin, which suggested that the abundance of HMGB1 released from the brain after ICH impeded the absorption of Hb leaked from the hematoma. Since anti-HMGB1 mAb neutralized extracellular HMGB1 in the previous study and inhibited the hemosiderin and iron deposition, it seems likely that treatment with anti-HMGB1 mAb facilitated the scavenging and clearance of Hb through the neutralization of HMGB1. To the best of our knowledge, this is the first study to examine the effects of humanized anti-HMGB1 in ICH injury in a non-human primate. Further studies in non-primate models may reveal the extensibility of our present results to humans. 

There is increasing evidence that suggests the usefulness of HMGB1-targeting therapy for the acute phase of ICH in animal models [22,23,32]. There are two HMGB1 inhibitors, glycyrrhizin and ethyl pyruvate (EP), and both have been suggested to have beneficial effects in rat ICH models [22,23,24]. Glycyrrhizin binds directly to both HMGB1 boxes, exerting protective effects against experimental TBI by interfering with the HMGB1-RAGE interaction [33,34]. Ohnishi et al. reported that intraperitoneal administration of glycyrrhizin attenuated ICH-induced edema and improved behavioral performance in a rat ICH model [22]. EP, a pyruvate derivative, is derived from the endogenous metabolite pyruvic acid [35]. EP acts through an HMGB1/TLR4/NF-κB-mediated pathway to inhibit HMGB1 expression in TBI rats [24]. Administration of EP has been shown to decrease the levels of HMGB1 and microglia activation around the hematoma after ICH in rats [23]. An in vitro experiment showed that EP inhibits the release of HMGB1 in LPS-stimulated RAW264.7 cells by deacetylation of HMGB1 via regulation of the SIRT/STAT pathways [36]. In our previous study, rat anti-HMGB1 mAb directly neutralized HMGB1, decreased the activated microglia and astrocytes, suppressed the expression of inflammation-related factors, and finally improved neurological function after ICH in rats [11]. In the present study, humanized anti-HMGB1 mAb also efficiently suppressed the secondary brain injury in marmoset ICH. 

After ICH, blood extravasation may release coagulation and fibrinolysis factors such as thrombin, fibrinogen, and plasmin. Moreover, red blood cells in the clot, as well as in the extravascular space, may result in hemolysis and subsequent release of Hb and its degradation products. All these factors have toxic effects on the brain cells and play an essential role in amplifying sterile inflammation in the presence of released HMGB1. For instance, Ito et al. reported that HMGB1 synergistically enhanced thrombin-induced coagulation and inflammation, leading to multiple organ failures in a disseminated intravascular coagulation model in rats [21]. Lin et al. demonstrated that Hb and HMGB1 acted synergistically to promote the release of proinflammatory cytokines from macrophages [20]. Hb was also reported to cause strong vasospasm [37]. Our previous data showed that the expressions of vasoconstriction-related receptors AT-1 PAR-1, V1 and TxA2 were up-regulated at 24 h after ICH and the up-regulation was suppressed by anti-HMGB1 mAb treatment [11]. 

Iron accumulation in the brain can stimulate free radical formation, leading to neuronal death, brain edema, and neurobehavioral deficits during ICH [38]. Ferrous iron generated during heme degradation can produce highly reactive hydroxyl radicals through Fenton reaction [39]. It should be noted that iron can induce neuronal death even after binding to ferritin because it can be locally released after being reduced to a ferrous form under acidic pH conditions during ICH [40,41]. Therefore, removing the hematoma and extravascular blood inside the brain may reduce the levels of toxic blood components [42]. In addition to surgical evacuation, there are three biological pathways involved in the removal of hematoma and Hb degradation products [43]: (1) erythrocytes are directly phagocytosed by activated macrophages/microglia through CD36; (2) Hb binds to haptoglobin to form a complex, which is endocytosed by macrophages/microglia via CD163; (3) heme binds to hemopexin and the resultant complex is transported into macrophages/microglia via CD91. Therefore, the efficient scavenging and clearance of Hb and its degradation products are necessary to minimize the release of proinflammatory mediators and free radicals that are toxic to neighboring cells and can lead to secondary brain injury [44]. Interestingly, a study on mouse ICH indicated that there was a robust increase in haptoglobin expression in both peripheral blood and the perihematomal zone following ICH [45]. The same study showed that haptoglobin was synthesized by oligodendroglia and released into the extracellular space to neutralize Hb toxicity and protect brain cells [45]. Another study discovered that haptoglobin also bound HMGB1 through a CD163-dependent pathway that confers protection against HMGB1-mediated inflammation in sepsis [46]. The brain subjected to ICH exhibits both Hb release and HMGB1 release, and these releases could mutually amplify the brain inflammation. Therefore, humanized anti-HMGB1 mAb could neutralize HMGB1 and liberate an abundant amount of haptoglobin to increase the clearance of Hb from hematoma clots. A clinical trial showed that, although surgical evacuation of hematoma decreased hematoma-related neurotoxicity, it failed to provide definitive neurological improvements in ICH patients [47]. In the present study, we clearly demonstrated that humanized anti-HMGB1 mAb not only inhibited inflammation, but also promoted hematoma clearance following ICH in marmosets (Figure 8). 

Free radicals are generated by blood cells or plasma-derived products, including free iron, heme and thrombin. Brain residual inflammatory cell microglia and infiltrating neutrophils also contribute to the production of ROS following ICH [48]. 4-hydroxynonenal (4-HNE) is a major aldehyde produced by lipid peroxidation in the membrane under oxidative stress, and diffusible 4-HNE is involved in the pathogenesis of several inflammation-related diseases [26,49,50,51] through the adduct formation with proteins, nucleic acids and lipids. A recent study also revealed that 4-HNE- stimulated phosphatidylserine externalization in erythrocytes led to erythrocytes aggregation and hemolysis in association with endothelial cell dysfunction [51]. In this study, we observed a marked increase in 4-HNE in the hematoma brain and bloodstream following ICH. These findings suggest that erythrocyte lysis in the presence of Hb, heme, and iron facilitated 4-HNE accumulation, which in turn promoted erythrocyte lysis as a vicious cycle in the hemorrhaged brain. Lee et al. reported that 4-HNE treatment enlarged the cerebral infarct area and increased oxidative damage in MCAO rats [26]. Our data show that a massive accumulation of 4-HNE persisted in the hematoma zone even 12 d after ICH onset, which manifested as continuous lipid peroxidation following ICH. However, humanized anti-HMGB1 mAb probably reduced lipid peroxidation in both the acute and subacute phases, as observed by the decreased levels of 4-HNE in the brain and the plasma. Together, these results suggest that humanized anti-HMGB1 mAb may effectively suppress ROS-induced tissue damage and erythrocyte lysis by a decrease in 4-HNE accumulation.

Up to 14% and 6% of patients with ICH experience early seizures and delayed seizures following ICH, respectively [52,53]. This means that hemorrhage-induced brain injuries may cause epileptogenesis through the associated inflammatory responses and the formation of recurrent excitatory pathways leading to subacute and chronic seizures. Therefore, the inhibition of acute inflammation during ICH and the facilitation of the scavenging of Hb-derived molecules by anti-HMGB1 therapy should help to suppress the subsequent seizure onset. Maroso et al. reported another finding in support of the effectiveness of anti-HMGB1 treatment. Namely, they found that kainic acid-induced seizures in mice were enhanced by the pre-application of recombinant HMGB1 into the hippocampus through the HMGB1-TLR4 axis [54]. In other studies, rat anti-HMGB1 mAb inhibited the pathogenesis of epilepsy by preventing BBB permeability and inhibited the inflammatory process in pilocarpine-induced epilepsy [16,55]. Moreover, it was reported that anti-HMGB1 mAb suppressed the seizures induced by electroshock, kainate or pentylenetetrazole. In other words, the administration of anti-HMGB1 can not only protect the brain from ICH-induced injury but can also prevent or inhibit post-ICH seizures. 

In our previous study, we also reported that anti-HMGB1 mAb therapy conferred protection in an ischemic stroke model in rats by protecting the BBB and inhibiting the associated inflammatory responses in the rat brain. Although, tissue plasminogen activator (t-PA) administered within 4.5 h after ischemia onset was safe and effective in patients with ischemic brain injury [56], the major side effect of t-PA administration was a hemorrhagic transformation of ischemic stroke [57]. Therefore, anti-HMGB1 has been demonstrated to have beneficial effects on both ischemic and hemorrhagic stroke [11,12,58].

There are limitations to our study. (1) The ICH model was established by local injection of collagenase IV into the striatum in healthy marmosets. In the case of clinical settings, hypertension usually preexists in spontaneous ICH patients. (2) Even though our previous study demonstrated that the treatment time window of rat anti-HMGB1 mAb is at least 3 h after ICH [11], humanized anti-HMGB1 mAb was managed within 10 min after marmoset ICH onset. In future studies, it will be important to explore the time window of humanized anti-HMGB1 mAb in ICH marmosets.

In the present study, we clearly demonstrated that anti-HMGB1 mAb inhibited the deposition of hemosiderin and iron in the area surrounding the hematoma in ICH marmosets. These products probably originated from Hb due to the hemolysis of red blood cells in the bleeding sites. Thus, the treatment with anti-HMGB1 mAb appeared to facilitate the scavenging of Hb and its degradation products and reduced the secondary inflammatory responses enhanced and mediated by HMGB1.

## 5. Conclusions

In conclusion, our study revealed that acute intravenous injection of humanized an-ti-HMGB1 mAb produced a beneficial effect through the inhibition of the inflammatory responses, facilitation of the resolution of hematoma, and suppression of iron deposition after ICH. Treatment with anti-HMGB1 mAb could be a novel therapeutic strategy for ICH.

## Figures and Tables

**Figure 1 cells-11-02970-f001:**
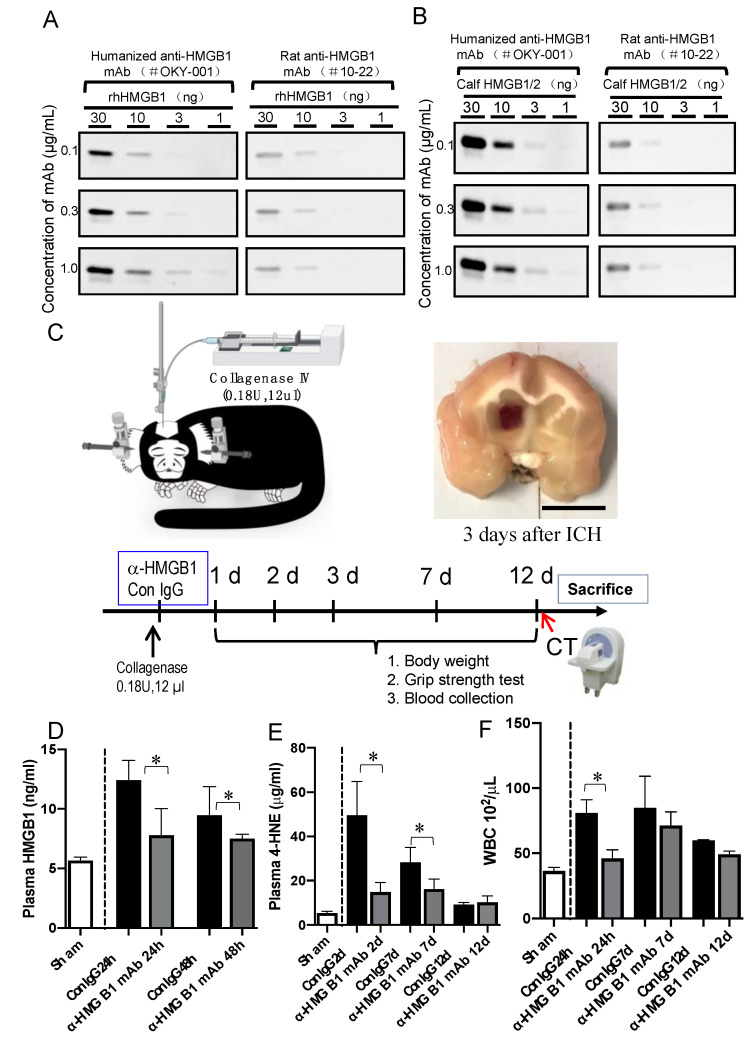
Experimental protocol for marmoset ICH. (**A**) Recognition of recombinant humanized-HMGB1 (rhHMGB1) by humanized anti-HMGB1 mAb (OKY-001). Different amounts of recombinant and native HMGB1 were treated with reducing sample buffer and electrophoresed. After transfer onto a PVDF membrane, HMGB1 was detected by OKY-001 or rat anti-HMGB1 original mAb (#10–22). (**B**) Recognition of native calf HMGB1/2 by humanized anti-HMGB1 mAb (OKY-001). Different amounts of native calf HMGB1/2 were treated with reducing sample buffer and electrophoresed. After transfer onto a PVDF membrane, HMGB1 was detected by OKY-001 or rat anti-HMGB1 original mAb (#10–22). (**C**) Schematic overview demonstrating the experimental strategy for the induction of local intracerebral hemorrhage with collagenase IV (0.18 U, 12 µL) in the common marmoset. After collagenase injection, humanized anti-HMGB1 (OKY-001) or the IgG control was injected i.v. from the tail vein. The representative image shows a section of the marmoset brain taken from d 3 after ICH. (**D**) Determination of plasma levels of HMGB1 by ELISA in a marmoset with ICH. Blood samples were collected before ICH and at 24, 48, and 12 d after h ICH induction. Values represent the means ± SE. * *p* < 0.05 compared with the IgG control. (**E**) Determination of plasma levels of 4-HNE adducts by ELISA in marmoset with ICH. Blood samples were collected before ICH and at 24 h, 7 d, and 12 d after ICH induction. Values represent the means ± SE. * *p* < 0.05 compared with the IgG control. (**F**) Whole blood WBC was counted before ICH and at 24 h and 7 and 12 d after ICH induction. Values represent the means ± SE. * *p* < 0.05 compared with the IgG control.

**Figure 2 cells-11-02970-f002:**
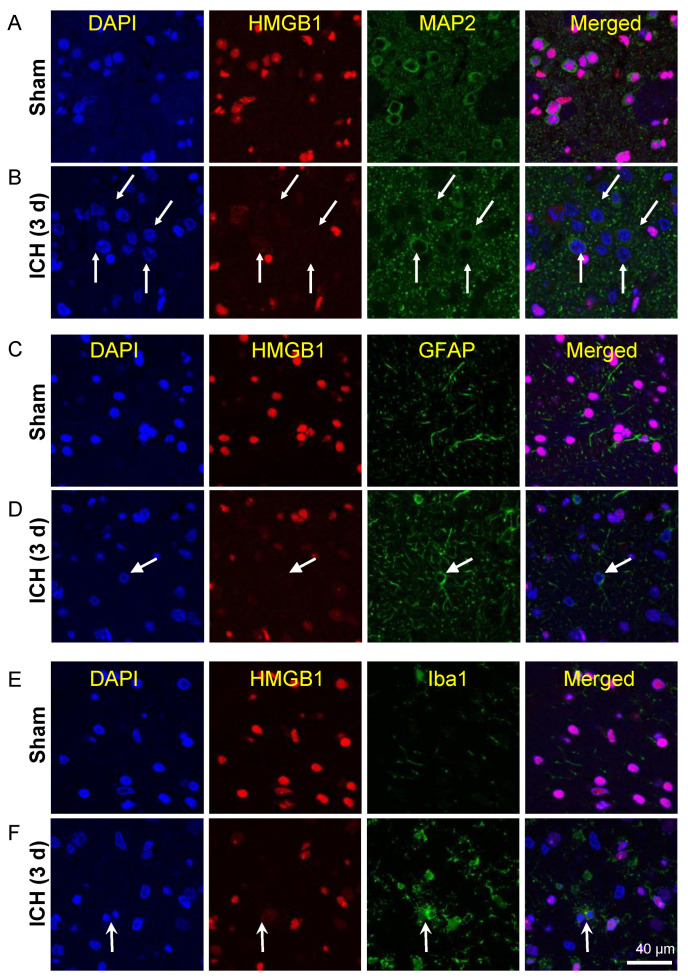
Release of HMGB1 in neurons, astrocytes, and microglia 3 d after ICH. (**A**) The neurons in the striatum were double-immunostained with anti-HMGB1 (red) and anti-MAP2 (green) in the brains of the sham animals. (**B**) The neurons in the peri-hematoma zone were double-immunostained with anti-HMGB1 (red) and anti-MAP2 (green) 3 d after ICH. The arrow indicates the HMGB1-negative neurons in an untreated marmoset. (**C**) The astrocytes in the striatum were double-immunostained with anti-HMGB1 (red) and anti-GFAP (green) in the brains of the sham-treated animals. (**D**) The astrocytes in the peri-hematoma zone were double-immunostained with anti-HMGB1 (red) and anti-GFAP (green) 3 d after ICH. The arrow indicates the HMGB1-negative neurons in an untreated marmoset. (**E**) The microglia in the stratum were double-immunostained with anti-HMGB1 (red) and anti-iba1 (green) in the brains of the sham animals. (**F**) The microglia in the peri-hematoma zone were double-immunostained with anti-HMGB1 (red) and anti-iba1 (green) 3 d after ICH. The arrow indicates the HMGB1-negative neurons in an untreated marmoset. The scale bar represents 40 μm.

**Figure 3 cells-11-02970-f003:**
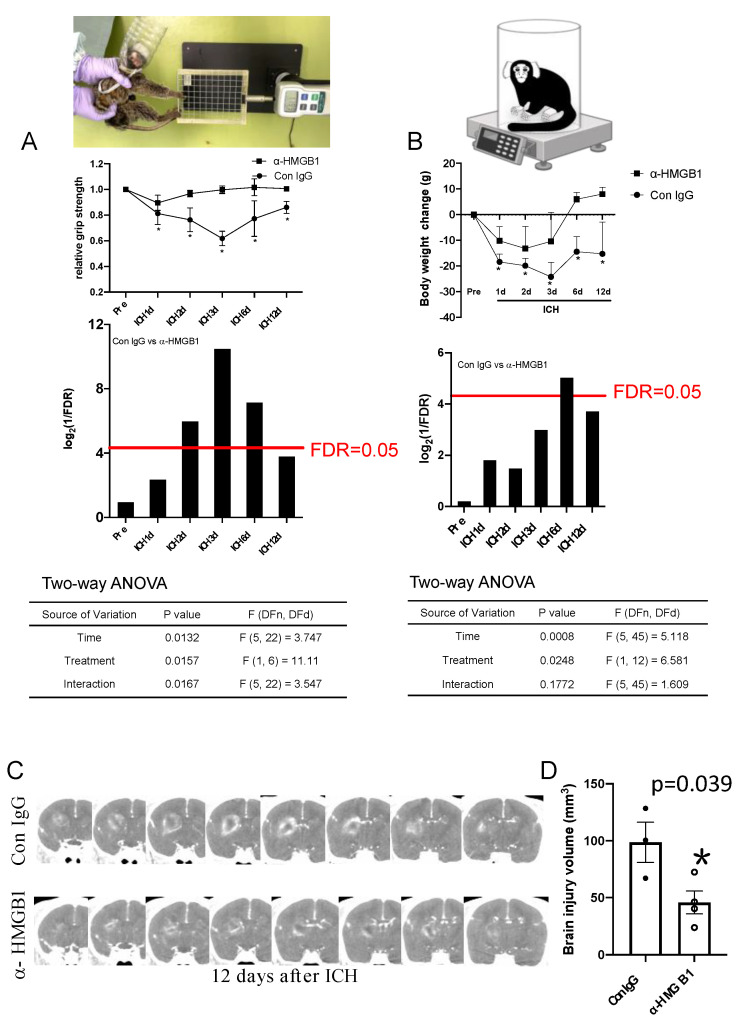
The determination of neurological deficit, body weight change and detection of brain injury by CT imaging. (**A**) An image of a marmoset griping the net. The force at which the marmoset releases the net was measured before and at 1, 2, 3, 6 and 12 d after ICH. The average values were determined from 5 trials. The relative grip strength was analyzed. Values represent the means ± SE. * *p* < 0.05 compared with the pre-ICH value. After a two-way ANOVA test, multiple testing with a false discovery rate (FDR) control for the relative grip strength curves was performed and the transformed FDR values were plotted (high plot bar with lower original FDR value). The red line represents *p* = 0.05. Values below the red line represent *p* < 0.05. ANOVA, analysis of variance. (**B**) Body weight was measured before and at 1, 2, 3, 6 and 12 d after ICH. The body weight change was analyzed between the IgG control and anti-HMGB1 groups. Values represent the means ± SE. * *p* < 0.05 compared with pre-ICH values. After a two-way ANOVA test, multiple testing with false discovery rate (FDR) control for the body weight change curves was performed and the transformed FDR values were plotted (high plot bar with lower original FDR value). The red line represents *p* = 0.05. Values below the red line represent *p* < 0.05. ANOVA, analysis of variance. (**C**) A series of computerized tomography (CT) images from the IgG control and anti-HMGB1 groups at 12 d after ICH. The whitish areas indicate the leakage of the contrast medium. The darkly colored area in the hematoma center also represents an injury area according to the HE staining result, even though the leakage of contrast media is undetectable in the center area. A representative image from 3–4 animals in each group is shown. (**D**) Quantification of brain injury volume determined as the outer boundaries of the contrast medium-leaking area in the IgG control and anti-HMGB1 groups. The whole volume contained a whitish area and a dark center area. * *p* < 0.05 compared with Con IgG values.

**Figure 4 cells-11-02970-f004:**
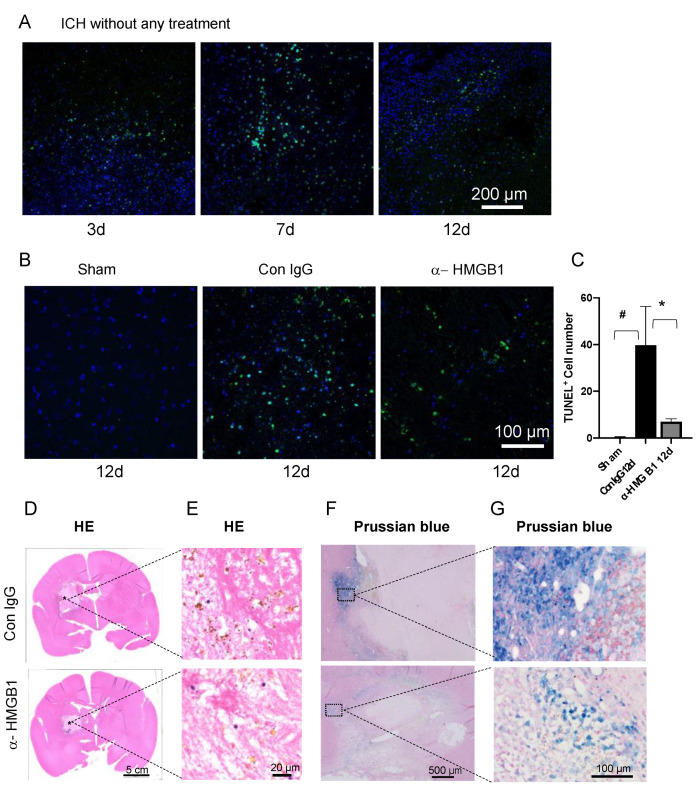
Detection of apoptotic cells and iron deposition after ICH. (**A**) The apoptotic cells were identified by TUNEL staining. Representative images of TUNEL staining at the border of the ipsilateral side of the brain at 3, 7, and 12 d after ICH without any treatment. The photos were taken at the border of the hematoma. Bar = 200 µm. (**B**) Representative images of TUNEL staining in sham, IgG control-treated, and anti-HMGB1 mAb-treated groups at 12 d after ICH. The pictures were taken at the border of the hematoma. Bar = 100 µm. (**C**) Quantification of the number of TUNEL-positive cells in the whole image field (0.16 cm^2^). Values represent the means ± SE. # *p* < 0.05 compared with the sham animals. * *p* < 0.05 compared with the IgG control. (**D**) Typical HE-staining images in the IgG control and anti-HMGB1 group at 12 d after ICH. (**E**) Deposition of hemosiderin (brown deposits), a hemoglobin breakdown product, was observed in both the IgG control-treated and anti-HMGB1-treated groups (the position is indicated by an asterisk in panel (**D**)). The anti-HMGB1 group showed a decreased amount of hemosiderin deposition. (**F**) Representative images of Prussian blue staining at the border of the hematoma in the IgG control-treated and anti-HMGB1-treated groups. The blue color indicates the accumulation of ferric iron. Bar = 500 µm. (**G**) Enlargement of the position of the square in panel F. Bar = 100 µm.

**Figure 5 cells-11-02970-f005:**
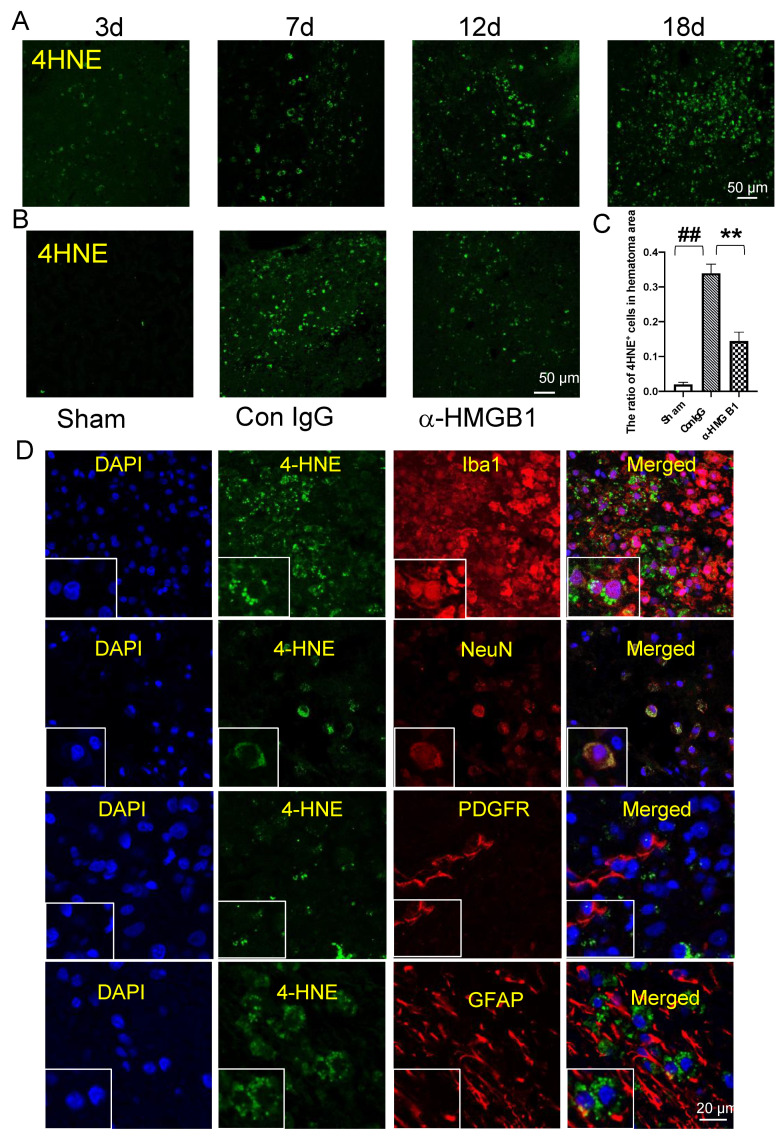
Detection of 4-HNE as an index of oxidative stress in the marmoset brain after ICH. (**A**) 4-HNE detection on the ipsilateral side of the brain at 3, 7, 12 and 18 d after ICH in an IgG control-treated marmoset. The pictures were taken at the border of the hematoma. Bar = 50 µm. (**B**) Representative images of 4-HNE detection on the ipsilateral side of the brain in the sham-treated, IgG control-treated and anti-HMGB1-treated animals at 12 d after ICH. Bar = 50 µm. (**C**) The quantification of the ratio of 4-HNE-positive cells to total DAPI-positive cells. Values represent the means ± SE. ## *p* < 0.01 compared with the sham-treatment group. ** *p* < 0.05 compared with the IgG control. (**D**) Double immunostaining of 4-HNE with Iba1, NeuN, PDGFR, or GFAP on the ipsilateral side of the brain at 7 d after ICH. The pictures were taken at the border of the hematoma. Bar = 20 µm. (**E**) Double immunostaining of 4-HNE with PDGFR, GFAP, or Iba1 on the ipsilateral side of the brain at 7 d after ICH. The representative images show the longitudinal section of one vessel. The pictures were taken in the peri-hematoma zone. Bar = 20 µm. (**F**) Double immunostaining of 4-HNE with PDGFR, GFAP, or Iba1 on the ipsilateral side of the brain at 7 d after ICH. The representative images show the section of one vessel. Bar = 20 µm. (**G**) Representative images of immunostaining with 4-HNE, ibal1, and GFAP on the contralateral side of the brain at 7 d after ICH. Bar = 20 µm.

**Figure 6 cells-11-02970-f006:**
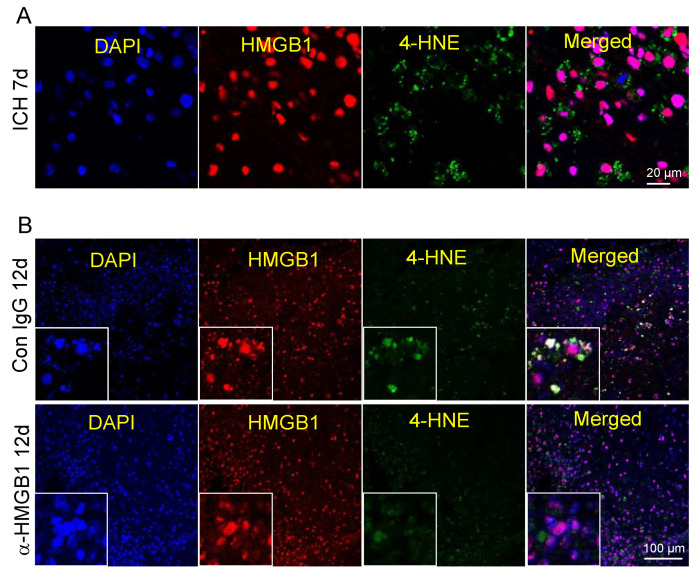
Immunostaining with 4-HNE and HMGB1 in the peri-hematoma zone. (**A**) Representative images of immunostaining with 4-HNE and HMGB1 on the ipsilateral side of the brain at 7 d after ICH in an IgG control-treated marmoset. The pictures were taken in the peri-hematoma zone. (**B**) Representative images of immunostaining with 4-HNE and HMGB1 on the ipsilateral side of the brain in the IgG control group and anti-HMGB1-treated group at 12 d after ICH. The photos were taken in the peri-hematoma zone. Bar = 100 µm.

**Figure 7 cells-11-02970-f007:**
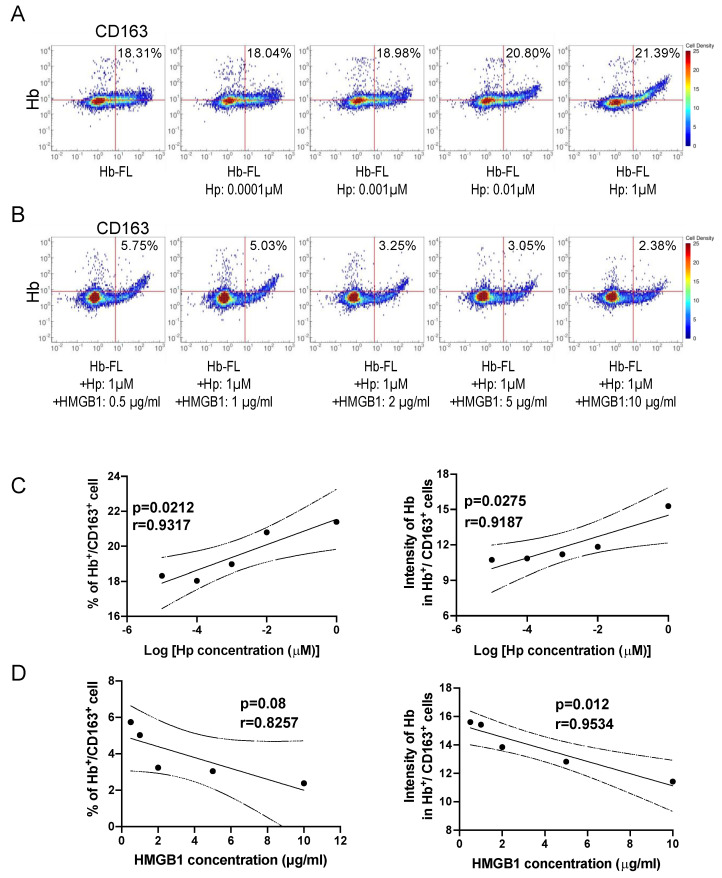
Flow cytometric analysis of the effect of haptoglobin-dependent uptake of Hb by recombinant HMGB1. (**A**) Dexamethasone-pretreated THP1 cells were incubated with alex647-Hb (1 µM) and different doses of haptoglobin (0, 0.0001, 0.001, 0.01 and 1 µM). The uptake of fluorescence-labeled Hb (alex647-Hb) into THP-1 cells increased in a haptoglobin dose-dependent manner. (**B**) Dexamethasone-pretreated THP1 cells were incubated with alex647-Hb (1 µM), haptoglobin (1 µM) and different doses of HMGB1 (0.5, 1, 2, 5, and 10 µg/mL). The uptake of fluorescence-labeled Hb (alex647-Hb) into THP-1 cells was decreased in an HMGB1 dose-dependent manner. (**C**) A plot showing a linear correlation between Hp concentration and the percentage of Hb^+^/CD163^+^ cells; and a linear correlation between Hp concentration and the intensity of Hb in Hb^+^/CD163^+^ cells in Figure 7A. The dash lines indicate the 95% confidence interval. (**D**) A plot showing a linear correlation between HMGB1 concentration and the percentage of Hb^+^/CD163^+^ cells; and a linear correlation between HMGB1 concentration and the intensity of Hb in Hb^+^/CD163^+^ cells in Figure 7B. The dash lines indicate the 95% confidence interval.

**Figure 8 cells-11-02970-f008:**
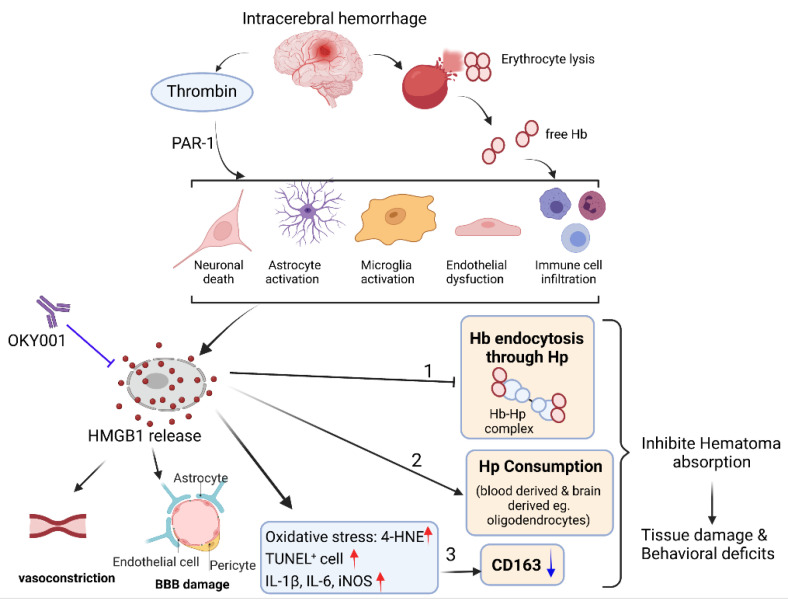
Proposed mechanism for the effect of anti-HMGB1 mAb in ICH. After intracerebral hemorrhage (ICH), blood is released into the brain, and erythrocyte lysis occurs, which causes the release of hemoglobin. Thrombin, free Hb, and the degradation of Hb (heme, iron) contribute to the pathological changes, including neuronal death, microglia activation, astrocyte activation, endothelial dysfunction, and immune cell infiltration, which lead to the release of HMGB1 from multiple sources. The released HMGB1 may play important roles in three aspects of Hb scavenging and clearance. Namely, the released HMGB1 could (1) consume the blood-derived, and brain-derived haptoglobin, (2) inhibit Hb endocytosis through competitive binding with haptoglobin, and (3) promote oxidative stress, which in turn may downregulate the expression of CD163. All of these events may inhibit hematoma absorption and finally lead to tissue damage and behavioral deficits. We drew this mechanism figure with BioRender.com (Biorender, Toronto, ON, Canada).

## Data Availability

The data that support the findings of this study are available from the corresponding author upon reasonable request.

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
