# Peer review of "Treatment of Marmoset Intracerebral Hemorrhage with Humanized Anti-HMGB1 mAb"

_cells, 2022, doi:10.3390/cells11192970_

Round 1

Reviewer 1 Report

The study "Treatment of marmoset intracerebral hemorrhage with humanized anti-HMGB1 mAb" represents indeed a high-quality research and there are no concerns or questions that need to be addressed. 

Author Response

Thank you very much for your review on our manuscript entitled “Treatment of marmoset intracerebral hemorrhage with humanized anti-HMGB1 mAb” by Wang et al. (Manuscript ID: cells-1904408). We are very glad to know that you recognized the significance of the findings in the manuscript. We sincerely appreciate your assistance in improving the manuscript.

Reviewer 2 Report

The authors performed a well structured study with humanized anti-HMGB1 mAB showing reduction of the release of HMGB1 from the brain into plasma resulting in reduction of 4HNE accumulate products. Furthermore, the authors showed that the antibody injection prevent the brain injury after ICH by various mechanisms. All in all, it is a very interesting study and I have only few minor questions: 

1. How many marmosets were included for this study? Since the ICH model was integrated first time to marmosets, how reliable is the striatal bleeding? Since the deposition of hemosiderin was analyzed, could it be an effect of a selection bias? 

2. Why did the authors performed CT for detection of brain injury? Is the contrast agent extravasation a standard tool for detection of brain injury? Why not MRI? 

Author Response

Thank you very much for your review and valuable comments on our manuscript entitled “Treatment of marmoset intracerebral hemorrhage with humanized anti-HMGB1 mAb” by Wang et al. (Manuscript ID: cells-1904408). We are very glad to know that you recognized the significance of the findings in the manuscript. We sincerely appreciate your assistance in improving the manuscript. In reply to your comments and suggestions, we modified the manuscript and have made appropriate revisions as described in the attached file.

Reviewer 3 Report

In this submitted manuscript, Wang et al. describe the effects of anti-HMGB1 in a primate model of ICH. Overall, the study was well done and the paper was well written. My concerns are as follows:

1) In the paper the authors write that the studies were performed in both sexes. Were sex differences analyzed and what were the results? If this analysis was done it should be reported.

2) The legend for Figure 1B describes ELISA data but in the figure is Western blot data. The legend needs to match the figure.

3) Why do the plasma levels of HMGB1 and 4-HNE go down over time in the IgG control group? Is the control IgG having a biological effect? This data should be discussed in the manuscript.

4) Section 3.2 should be moved to after section 3.4 and there is no mention of Figure 1F anywhere in the rest of the paper. This data should be discussed somewhere.

5) Figure 4A needs to be labeled like 4B. I don’t know what treatment groups I’m looking at in 4A.

6) PDGFR staining in 5D is weak and hard to see. A better example which clearly shows pericytes would be helpful.
